# Rectangular Flows for Manifold Learning

## Abstract

Normalizing flows allow for tractable maximum likelihood estimation of their parameters but are incapable of modelling low-dimensional manifold structure in observed data. Flows which injectively map from low- to high-dimensional space provide promise for fixing this issue, but the resulting likelihood-based objective becomes more challenging to evaluate. Current approaches avoid computing the entire objective – which may induce pathological behaviour – or assume the manifold structure is known beforehand and thus are not widely applicable. Instead, we propose two methods relying on tricks from automatic differentiation and numerical linear algebra to either evaluate or approximate the full likelihood objective, performing end-to-end manifold learning and density estimation. We study the trade-offs between our methods, demonstrate improved results over previous injective flows, and show promising results on out-of-distribution detection.

## 1. Introduction

Normalizing Flows (NFs) have recently become a staple of generative modelling, and particularly for density estimation (see Papamakarios et al. (2019) or Kobyzev et al. (2020) for a review). Here, we typically have access to a set of points in some high-dimensional space $\mathbb{R}^D$, which NFs model as the pushforward of some simple distribution on $\mathbb{R}^D$ through a parametrized bijection. Although this construction can admit tractable maximum likelihood training, the learned density has $D$-dimensional support; this directly contradicts the manifold hypothesis (Bengio et al., 2013) which states that high-dimensional data lives on a lower-dimensional manifold embedded in ambient space.

Instead, we may consider *injective* flows to circumvent this misspecification. These now map a random variable on

[1]Anonymous Institution, Anonymous City, Anonymous Region, Anonymous Country. Correspondence to: Anonymous Author <anon.email@domain.com>.

Preliminary work. Under review by INNF+ 2021. Do not distribute.

$\mathbb{R}^d$ into $\mathbb{R}^D$, defining a distribution on some $d$-dimensional manifold embedded in $\mathbb{R}^D$. We have access to a change-of-variable formula as in NFs, but the volume-change term now becomes much more difficult to evaluate. While there have been efforts towards training flows where the resulting distribution is supported on a low-dimensional manifold (e.g. (Rezende et al., 2020; Brehmer & Cranmer, 2020)), these approaches either assume the manifold is known beforehand or otherwise avoid the volume-change term. Both of these are undesirable: in the former, we generally do not know the manifold structure *a priori*; the latter can result in learning manifolds to which it is difficult to assign density.

In this work, we show that likelihood-based density estimation for injective flows can be made tractable. We propose two methods for backpropagating through the injective volume-change term which rely on careful application of forward- and backward-mode automatic differentiation. The first method involves exact evaluation of this term and its gradient which incurs a higher memory cost; the second uses conjugate gradients and Hutchinson's trace estimator to obtain unbiased stochastic gradient estimates. Unlike previous work, our methods do not need the data manifold to be specified beforehand, and simultaneously estimate this manifold along with the distribution on it end-to-end, thus enabling maximum likelihood training to occur. Ours are the first methods to backpropagate through the volume-change term in ambient dimensions $D$ approaching 1,000. We study the trade-off between memory and variance introduced by our methods and show improvements over injective flow baselines for density estimation. We also show that injective flows obtain state-of-the-art performance for likelihood-based Out-of-Distribution (OoD) detection.

## 2. Background and Motivation

### 2.1. Rectangular Normalizing Flows

Standard NFs unrealistically result in the learned density $p_X$ having $D$-dimensional support. To overcome this, we first follow Brehmer & Cranmer (2020), where an injective mapping $g_\phi : \mathbb{R}^d \to \mathbb{R}^D$ with $d < D$ is constructed. Here, $Z \in \mathbb{R}^d$ is the low-dimensional variable used to model the data as $X := g_\phi(Z)$. A well-known result from differential geometry (Krantz & Parks, 2008), provides a change-of-

variable formula for $x \in \mathcal{M}_\phi := \{g_\phi(z) : z \in \mathbb{R}^d\}$:

$$p_X(x) = p_Z(z_\phi) \left| \det \mathbf{J}[g_\phi]^\top(z_\phi)\mathbf{J}[g_\phi](z_\phi) \right|^{-1/2}, \quad (1)$$

where $z_\phi := g_\phi^{-1}(x)$, and $p_X(x) = 0$ for $x \notin \mathcal{M}_\phi$. The Jacobian-transpose-Jacobian (JtJ) determinant now characterizes the change in volume from $Z$ to $X$. We make several relevant observations: $(i)$ The Jacobian matrix $\mathbf{J}[g_\phi](g_\phi^{-1}(x)) \in \mathbb{R}^{D \times d}$ is no longer a square matrix, and we thus refer to these flows as *rectangular*. $(ii)$ $g_\phi^{-1} : \mathcal{M}_\phi \to \mathbb{R}^d$ is only properly defined on $\mathcal{M}_\phi$ and not $\mathbb{R}^D$, and $p_X$ is now supported on the $d$-dimensional manifold $\mathcal{M}_\phi$. $(iii)$ This is *not* a density with respect to the Lebesgue measure; the dominating measure is the Riemannian measure on the manifold $\mathcal{M}_\phi$ (Pennec, 2006). $(iv)$ When $d = D$, we recover the standard change-of-variable.

Since data points $x$ will almost surely not lie exactly on $\mathcal{M}_\phi$, we use a left inverse $g_\phi^\dagger : \mathbb{R}^D \to \mathbb{R}^d$ such that $g_\phi^\dagger(g_\phi(z)) = z$ for all $z \in \mathbb{R}^d$ in place of $g_\phi^{-1}$. This exists by injectivity and is properly defined on $\mathbb{R}^D$, unlike $g_\phi^{-1}$ which only exists on $\mathcal{M}_\phi$. Setting $z_\phi := g_\phi^\dagger(x)$ in (1) is equivalent to projecting $x$ onto $\mathcal{M}_\phi$ as $x \leftarrow g_\phi(g_\phi^\dagger(x))$, and then evaluating the density from (1) at the projected point.

Now, $g_\phi$ is injectively constructed as follows:

$$g_\phi = \tilde{f}_\theta \circ \mathtt{pad} \circ h_\eta \quad \text{and} \quad g_\phi^\dagger = h_\eta^{-1} \circ \mathtt{pad}^\dagger \circ \tilde{f}_\theta^{-1}, \quad (2)$$

where $\tilde{f}_\theta : \mathbb{R}^D \to \mathbb{R}^D$ and $h_\eta : \mathbb{R}^d \to \mathbb{R}^d$ are both square flows, $\phi := (\theta, \eta)$, and $\mathtt{pad} : \mathbb{R}^d \to \mathbb{R}^D$ and $\mathtt{pad}^\dagger : \mathbb{R}^D \to \mathbb{R}^d$ are defined as $\mathtt{pad}(z) = (z, \mathbf{0})$ and $\mathtt{pad}^\dagger(z, z') = z$, where $\mathbf{0}, z' \in \mathbb{R}^{D-d}$. We can thus rewrite (1) using this specific form of $g_\phi$, with details in Appendix B.

Constructing flows with a tractable volume-change term is more challenging than in the standard case. Brehmer & Cranmer (2020) thus propose a two-step training procedure, wherein $f_\theta := \tilde{f}_\theta \circ \mathtt{pad}$ and $h_\eta$ are trained separately, to avoid this calculation. Training $f_\theta$ involves minimizing the reconstruction error $\|x - f_\theta(f_\theta^\dagger(x))\|_2^2$, which encourages the observed data to lie on $\mathcal{M}_\theta$. Then, since $h_\eta$ will not appear in the determinant term in (1), it can be taken to be any $d$-dimensional NF and, fixing $\theta$, $\eta$ can be learned via maximizing its likelihood over the lower-dimensional points $\{f_\theta^\dagger(x_i)\}_i$. In practice, gradients steps in $\theta$ and $\eta$ are alternated. This procedure circumvents evaluation of the JtJ term, but we soon show that this comes with downsides.

### 2.2. Motivation

**Dimensionality issues** Problems originating from dimensionality mismatch have been observed throughout the deep generative modelling literature. Dai & Wipf (2019) show that using powerful variational autoencoders supported on $\mathbb{R}^D$ to model data living in a low-dimensional manifold results in the manifold itself being learned, but not the distribution on it. Cornish et al. (2020) demonstrate the drawbacks of using normalizing flows for estimating the density of topologically-complex data, but still model the support as being $D$-dimensional; Behrmann et al. (2021) provide a related result characterizing non-invertibility of trained flows. This body of work strongly motivates the development of models whose support has matching topology – including dimension – to that of the true data distribution.

**Manifold flows** A challenge to overcome for NFs on manifolds is the JtJ term; this is currently handled in one of two ways. The first assumes the manifold is known beforehand (Rezende et al., 2020), limiting its general applicability to low-dimensional data where the true manifold can be known. The second group circumvents the computation of this term entirely; this includes the aforementioned Brehmer & Cranmer (2020). Kumar et al. (2020) use a loose lower bound of the log-likelihood and do not explicitly enforce injectivity, so that the change-of-variable almost surely does not hold. Cunningham et al. (2020) propose to convolve the manifold distribution with Gaussian noise, which results in the model having high-dimensional support.

**Why optimize this term?** We can imagine a situation where, even if $f_\theta$ maps to the correct manifold, it might unnecessarily change volume in such a way that makes correctly learning $h_\eta$ more challenging than it needs to be. For example, there is nothing in the two-step objective encouraging $f_\theta$ to learn a manifold parametrized with a well-controlled "speed", which we observe to be an issue in Figure 2 of the experiments. This is but one example of a failure which could have been avoided by learning the manifold in a density-aware fashion including the JtJ term.

## 3. Rectangular Flow Maximum Likelihood

### 3.1. Our Optimization Objective

We have noted that including JtJ in the optimization is sensible, but (1) corresponds to the density of the projection of $x$ onto $\mathcal{M}_\theta$. Thus, optimizing only this would not result in learning $\mathcal{M}_\theta$ such that observed data lies on it, only encouraging projected data points to have high likelihood. Instead, we use the KKT conditions (Karush, 1939; Kuhn & Tucker, 1951) to maximize the following Lagrangian in $\phi$ subject to the constraint that the reconstruction error should be smaller than some threshold:

$$\log p_Z(z_\phi) - \log |\det \mathbf{J}[h_\eta](z_\phi)| \quad (3)$$
$$- \frac{1}{2}\log \det J_\theta^\top(x)J_\theta(x) - \beta \left\|x - f_\theta\left(f_\theta^\dagger(x)\right)\right\|_2^2,$$

where we treat $\beta > 0$ as a hyperparameter, and denote $\mathbf{J}[f_\theta](f_\theta^\dagger(x))$ as $J_\theta(x)$ and again $z_\phi := g_\phi^\dagger(x)$ for simplicity. We now make a technical but relevant observation about our objective: since our likelihoods are Radon-Nikodym derivatives with respect to the Riemannian measure on $\mathcal{M}_\theta$, different values of $\theta$ will result in different dominating measures. One should thus be careful to compare likelihoods for models with different values of $\theta$. However, thanks to the smoothness of the objective over $\theta$, we should expect likelihoods for values of $\theta$ which are "close enough" to be comparable for practical purposes. In other words, comparisons remain reasonable locally, and the gradient of the volume-change term still contains information that helps learning $\mathcal{M}_\theta$ in such a way that $h_\eta$ can easily learn a density on the pulled-back dataset $\{f_\theta^\dagger(x_i)\}_i$.

### 3.2. Optimizing our Objective: Stochastic Gradients

All the terms in (3) are straightforward to evaluate and back-propagate through except for the third one; in this section we show how to obtain unbiased stochastic estimates of its gradient. We now drop the dependence of the Jacobian on $x$ from our notation and write $J_\theta$, knowing that the end computation will be parallelized over a batch $\{x_i\}_i$. We assume access to an efficient matrix-vector product routine, i.e. computing $J_\theta^\top J_\theta \epsilon$ can be quickly achieved for any $\epsilon \in \mathbb{R}^d$. We elaborate on how we obtain these matrix-vector products in the next section. It is a well known fact from matrix calculus (Petersen & Pedersen, 2008) that:

$$\frac{\partial}{\partial \theta_j} \log \det J_\theta^\top J_\theta = \mathrm{tr}\left((J_\theta^\top J_\theta)^{-1} \frac{\partial}{\partial \theta_j} J_\theta^\top J_\theta\right), \quad (4)$$

where $\mathrm{tr}$ denotes the trace operator and $\theta_j$ is the $j$-th element of $\theta$. Next, we use Hutchinson's trace estimator (Hutchinson, 1989), which says that for any matrix $M \in \mathbb{R}^{d \times d}$, $\mathrm{tr}(M) = \mathbb{E}_\epsilon[\epsilon^\top M \epsilon]$ for any $\mathbb{R}^d$-valued random variable $\epsilon$ with zero mean and identity covariance matrix. We can thus obtain an unbiased stochastic estimate of our gradient as:

$$\frac{\partial}{\partial \theta_j} \mathrm{logdet}\, J_\theta^\top J_\theta \approx \frac{1}{K} \sum_{k=1}^{K} \epsilon_k^\top (J_\theta^\top J_\theta)^{-1} \frac{\partial}{\partial \theta_j} J_\theta^\top J_\theta \epsilon_k, \quad (5)$$

where $\{\epsilon_k\}_k$ are typically sampled either from standard Gaussian or Rademacher distributions. Naïve computation of the above estimate remains intractable without explicitly constructing $J_\theta^\top J_\theta$. Fortunately, the $J_\theta^\top J_\theta \epsilon$ terms can be trivially obtained using the given matrix-vector product routine, avoiding the construction of $J_\theta^\top J_\theta$, and then $\partial/\partial\theta_j J_\theta^\top J_\theta \epsilon$ follows by taking the gradient w.r.t. $\theta$.

Yet there is still the issue of computing $\epsilon^\top (J_\theta^\top J_\theta)^{-1} = [(J_\theta^\top J_\theta)^{-1} \epsilon]^\top$. We use Conjugate Gradients (CG) (Nocedal & Wright, 2006) in order to achieve this. CG is an iterative method to solve problems of the form $Au = \epsilon$ for given

$A \in \mathbb{R}^{d \times d}$ (in our case $A = J_\theta^\top J_\theta$) and $\epsilon \in \mathbb{R}^d$; we include the CG algorithm in Appendix D for completeness. CG has several important properties. First, it is known to recover the solution (assuming exact arithmetic) after at most $d$ steps, which means we can evaluate $A^{-1}\epsilon$. The solution converges exponentially (in the number of iterations $\tau$) to the true value (Shewchuk et al., 1994), so often $\tau \ll d$ iterations are sufficient for accuracy to many decimal places. Second, CG only requires a method to compute matrix-vector products against $A$, and does not require access to $A$ itself. One such product is performed at each iteration, and CG thus requires at most $d$ of these products, although again $\tau \ll d$ product usually suffice. This results in $\mathcal{O}(\tau d^2)$ solve complexity—less than the $\mathcal{O}(d^3)$ required by direct inversion. We denote $A^{-1}\epsilon$ computed with conjugate gradients as $\mathrm{CG}(A; \epsilon)$. We can then compute the estimator from (5) as:

$$\frac{\partial}{\partial \theta_j} \mathrm{logdet}\, J_\theta^\top J_\theta \approx \frac{1}{K} \sum_{k=1}^{K} \mathrm{CG}\left(J_\theta^\top J_\theta; \epsilon_k\right)^\top \frac{\partial}{\partial \theta_j} J_\theta^\top J_\theta \epsilon_k \quad (6)$$

In practice, we implement this term by applying a `stop_gradient` on the CG step, thereby allowing us to avoid implementing a custom backward pass. We add this term into (3) and write out in full the contribution of a point $x$ to the training objective in the Appendix (Equation (12)).

### 3.3. AD Considerations: The Exact Method and the Forward-Backward AD Trick

Here we derive the routine for vector products against $J_\theta^\top J_\theta$, along with an exact method that avoids Hutchinson's estimator but has increased memory requirements. We will use commonly-known properties of AD to derive our approach, which we review in Appendix E. First, consider the problem of explicitly constructing $J_\theta$. This construction can then be used to evaluate $J_\theta^\top J_\theta$ and exactly compute its log determinant, thus avoiding having to use stochastic gradients as in the previous section. We refer to this procedure as the *exact* method. Naïvely, one might try to explicitly construct $J_\theta$ using only backward-mode AD, which would require $D$ vector-Jacobian products (vjps) of the form $v^\top J_\theta$ – one per basis vector $v \in \mathbb{R}^D$. A better way to explicitly construct $J_\theta$ is with forward-mode AD, which only requires $d$ Jacobian-vector products (jvps) $J_\theta \epsilon$, again one per basis vector $\epsilon \in \mathbb{R}^d$. We use a custom implementation of forward-mode AD in the popular PyTorch (Paszke et al., 2019) library[1] for the exact method, as well as for the forward-backward AD trick described below.

We now explain how to combine forward- and backward-mode AD to obtain efficient matrix-vector products against

---

[1] PyTorch has a forward-mode AD implementation which relies on the "double backward" trick, which is known to be memory-inefficient. See https://j-towns.github.io/2017/06/12/A-new-trick.html for a description.

$J_\theta^\top J_\theta$ in order to obtain the tractable gradient estimates from the previous section. Note that $v := J_\theta \epsilon$ can be computed with a single `jvp` call, and then $J_\theta^\top J_\theta \epsilon = [v^\top J_\theta]^\top$ can be efficiently computed using only a `vjp` call. We refer to this way of computing matrix-vector products against $J_\theta^\top J_\theta$ as the *forward-backward AD trick*. Note that (12) requires $K(\tau + 1)$ such matrix-vector products, which might appear less efficient as it could be greater than the $d$ `jvp`s required by the exact method. However, the stochastic method is much more memory-efficient than its exact counterpart when optimizing: of the $K(\tau + 1)$ matrix-vector products needed to evaluate (12), only $K$ require gradients with respect to $\theta$. Thus only $K$ `jvp`s and $K$ `vjp`s, along with their intermediate steps, must be stored in memory over a training step. In contrast, the exact method requires gradients for every one of its $d$ `jvp` computations, which requires storing these along with their intermediate steps in memory.

Our proposed methods thus offer a memory-variance trade-off. Increasing $K$ in the stochastic method results in larger memory requirements which imply longer training times, as the batch size must be set lower. On the other hand, the larger the memory cost, the smaller the variance of the gradient. This still holds true for the exact method, which results in exact gradients, at the cost of increased memory requirements (as long as $K \ll d$; if $K$ is large enough the stochastic method should never be used over the exact one).

## 4. Experiments

We compare our methods against that of Brehmer & Cranmer (2020), and study the memory vs. variance trade-off. We include all experimental details in Appendix H. Figure 2 shows how the two-step method (TS) correctly recovers the manifold, but not the distribution on it when trying to learn a simple von Mises ground truth distribution on the unit circle in $\mathbb{R}^2$, which our method (ML) handily recovers.

We also compare the methods with the tabular datasets used by Papamakarios et al. (2017), along with MNIST and FMNIST. Due to space constraints, we include our results in Appendix A, where we show that: $(i)$ our maximum likelihood methods better recover the target distribution, as measured by FID score (Heusel et al., 2017); $(ii)$ our stochastic version with $K = 1$ is competitive against its more memory-hungry alternatives; and $(iii)$ rectangular flows

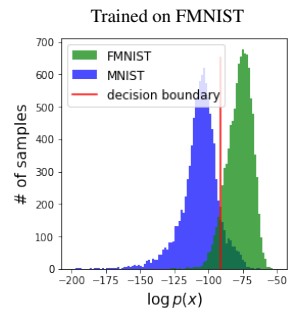

*Figure 1.* OoD detection with RNFs-ML (exact).

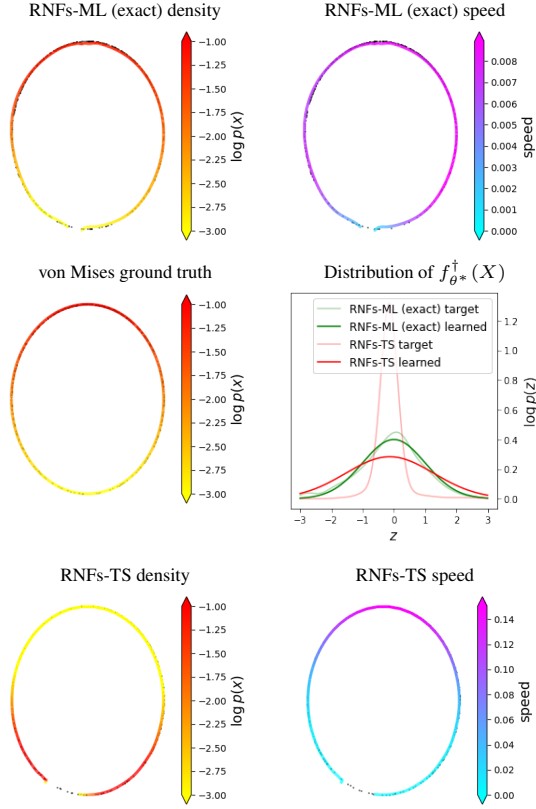

*Figure 2.* Left column: RNFs-ML (exact) (top), von Mises ground truth (middle), and RNF-TS (bottom). Right column: Speed at which $f_{\theta*}$ maps to $\mathcal{M}_{\theta*}$ (measured as $l_2$ distance between uniformly spaced consecutive points in $\mathbb{R}$ mapped through $f_{\theta*}$) for RNFs-ML (exact) (top), RNFs-TS (bottom), and distribution $h_\eta$ has to learn in order to recover the ground truth, fixing $\theta^*$ (middle). We can see that RNFs-ML map from low to high dimensions at a more constant speed, thus providing a simpler $z$ distribution for $h_\eta$ to learn. RNFs-TS map at a higher speed towards the top of the circle which impacts density estimates.

perform very well for OoD detection. In particular, they assign higher likelihoods to FMNIST than to MNIST when trained on the former, contrary to what has been observed in previous NFs literature (Nalisnick et al., 2019), as can be seen in Figure 1.

## 5. Conclusions

In this paper we argue for the importance of likelihood-based training of rectangular flows, and introduce two methods allowing to do so. We study the benefits of our methods, and empirically show that they are preferable to current alternatives. We anticipate improvements to our methods with more powerful flow architectures than RealNVP, along with advancements in specifying flow models with more flexible topological properties.

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

# A. Main Experimental Results

In all our experiments, w use the real NVP (Dinh et al., 2017) architecture for all flows, except we do not use batch normalization (Ioffe & Szegedy, 2015) as it causes issues with vjp computations. We point out that all comparisons remain fair, including a detailed explanation of this phenomenon in Appendix F, along with all experimental details in Appendix H. Throughout, we use the labels RNFs-ML for our maximum likelihood training method, RNFs-TS for the two-step method, and RNFs for rectangular NFs in general. For most runs, we found it useful to anneal the likelihood term(s). That is, at the beginning of training we optimize only the reconstruction term, and then slowly incorporate the other terms. This likelihood annealing procedure helped avoid local optima where the manifold is not recovered (large reconstruction error) but the likelihood of projected data is high.

## A.1. Simulated Data

We consider a simulated dataset where we have access to ground truth, which allows us to empirically verify the deficiencies of RNFs-TS. We use a von Mises distribution, which is supported on the one-dimensional unit circle in $\mathbb{R}^2$. Figure 2 shows this distribution, along with its estimates from RNFs-ML (exact) and RNFs-TS. As previously observed, RNFs-TS correctly approximate the manifold, but fail to learn the right distribution on it. In contrast we can see that RNFs-ML, by virtue of including the Jacobian-transpose-Jacobian term in the optimization, manage to recover both the manifold and the distribution on it (**top left panel**), while also resulting in an easier-to-learn low-dimensional distribution (**middle right panel**) thanks to $f_{\theta^*}$ mapping to $\mathcal{M}_{\theta^*}$ at a more consistent speed (**top right panel**). We do point out that, while the results presented here are representative of usual runs for both methods, we also had runs with different results which we include in Appendix H. We finish with the observation that even though the line and the circle are not homeomorphic and thus RNFs are not perfectly able to recover the support, they manage to adequately approximate it.

## A.2. Tabular Data

We now turn our attention to the tabular datasets used by Papamakarios et al. (2017), now a common benchmark for NFs as well. As previously mentioned, one should be careful when comparing models with different supports, as we cannot rely on test likelihoods as a metric. We take inspiration from the FID score (Heusel et al., 2017), which is commonly used to evaluate quality of generated images when likelihoods are not available. The FID score compares the first and second moments of a well-chosen statistic from the model and data distributions using the squared Wasserstein-2 metric (between Gaussians). Instead of using the last hidden layer of a pre-trained classifier as is often done for images, we take the statistic to be the data itself: in other words, our metric compares the mean and covariance of generated data against those of observed data with the same squared Wasserstein-2 metric. We include the mathematical formulas for computing both FID and our modified version for tabular data in Appendix G. We use early stopping with our FID-like score across all models. Our results are summarized in Table 1, where we can see that RNFs-ML consistently do a better job at recovering the underlying distribution. Once again, these results emphasize the benefits of including the Jacobian-transpose-Jacobian in the objective. Interestingly, except for HEPMASS, the results from our stochastic version with $K = 1$ are not significantly exceeded by the exact version or using a larger value of $K$, suggesting that the added variance does not result in decreased empirical performance. We highlight that no tuning was done (except on GAS for which we changed $d$ from 4 to 2), RNFs-ML outperforming RNFs-TS out-of-the-box here (details are in Appendix H). We report training times in Appendix H, and observe that RNFs-ML take a similar amount of time as RNFs-TS to train for datasets with lower values of $D$, and while we do take longer to train for the other datasets, our training times remain reasonable and we often require fewer epochs to converge.

## A.3. Image Data and Out-of-Distribution Detection

We also compare RNFs-ML to RNFs-TS for image modelling on MNIST and FMNIST. We point out that these datasets have ambient dimension $D = 784$, and being able to fit RNFs-ML is in itself noteworthy: to the best of our knowledge no previous method has scaled optimizing the Jacobian-transpose-Jacobian term to these dimensions. We use FID scores both for comparing models and for early stopping during training. We also used likelihood annealing and set $d := 20$, with all experimental details again given in Appendix H. We report FID scores in Table 2, where we can see that we outperform RNFs-TS. Our RNFs-ML ($K = 1$) variant also outperforms its decreased-variance counterparts. This is partially explained by the fact that we used this variant to tune RNFs-ML, but we also hypothesize that this added variance can be helpful because of the remaining (non-dimension-based) topological mismatch. Nonetheless, once again these results suggest that

Table 1. FID-like metric for tabular data (lower is better). Bolded runs are the best or overlap with it.

| Method | POWER | GAS | HEPMASS | MINIBOONE |
|---|---|---|---|---|
| RNFs-ML (exact) | **0.069 ± 0.014** | **0.138 ± 0.021** | **0.486 ± 0.032** | **0.978 ± 0.082** |
| RNFs-ML ($K = 1$) | **0.083 ± 0.015** | **0.110 ± 0.021** | 0.779 ± 0.191 | **1.001 ± 0.051** |
| RNFs-ML ($K = 10$) | 0.113 ± 0.037 | **0.140 ± 0.013** | **0.495 ± 0.055** | **0.878 ± 0.083** |
| RNFs-TS | 0.178 ± 0.021 | 0.161 ± 0.016 | 0.649 ± 0.081 | 1.085 ± 0.0622 |

Table 2. FID scores (lower is better) and decision stump OoD accuracy (higher is better).

| Method | FID | | OoD ACCURACY | |
|---|---|---|---|---|
| | MNIST | FMNIST | MNIST → FMNIST | FMNIST → MNIST |
| RNFs-ML (exact) | 36.09 | 296.01 | 92% | 91% |
| RNFs-ML ($K = 1$) | **33.98** | **288.39** | 97% | 78% |
| RNFs-ML ($K = 4$) | 42.90 | 342.91 | 77% | 89% |
| RNFs-TS | 35.52 | 318.59 | **98%** | **96%** |

the variance induced by our stochastic method is not empirically harmful. We also report training times in Appendix H, where we can see the computational benefits of our stochastic method.

We further evaluate the performance of RNFs for OoD detection. Nalisnick et al. (2019) pointed out that square NFs trained on FMNIST assign higher likelihoods to MNIST than they do to FMNIST. While there has been research attempting to fix this puzzling behaviour (Alemi et al., 2017; 2018; Choi et al., 2018; Ren et al., 2019), to the best of our knowledge no method has managed to correct it using only likelihoods of trained models. Figure 1 shows that RNFs remedy this phenomenon, and that models trained on FMNIST assign higher test likelihoods to FMNIST than to MNIST. This correction does not come at the cost of strange behaviour now emerging in the opposite direction (i.e. when training on MNIST, see Appendix H for a histogram). Table 2 quantifies these results (arrows point from in-distribution datasets to OoD ones) with the accuracy of a decision stump using only log-likelihood, and we can see that the best-performing RNFs models essentially solve this OoD task. While we leave a formal explanation of this result for future work, we believe this discovery highlights the importance of properly specifying models and of ensuring the use of appropriate inductive biases, in this case low intrinsic dimensionality of the observed data. We point out that this seems to be a property of RNFs, rather than of our ML training method, although our exact method is still used to compute these log-likelihoods at test time. We include additional results on OoD detection using reconstruction errors – along with a discussion – in Appendix H, where we found the opposite unexpected behaviour: FMNIST always has smaller reconstruction errors, regardless of which dataset was used for training.

## B. Injective Change-of-Variable Formula and Stacking Injective Flows

First, we note the density of projected points, and then we will derive it. Defining $f_\theta := \tilde{f}_\theta \circ \texttt{pad}$ and $f_\theta^\dagger := \texttt{pad}^\dagger \circ \tilde{f}_\theta^{-1}$, our construction of $g_\phi$ yields:

$$p_X(x) = p_Z\left(g_\phi^\dagger(x)\right) \left|\det \mathbf{J}[h_\eta]\left(g_\phi^\dagger(x)\right)\right|^{-1} \left|\det \mathbf{J}[f_\theta]^\top \left(f_\theta^\dagger(x)\right) \mathbf{J}[f_\theta]\left(f_\theta^\dagger(x)\right)\right|^{-1/2}. \tag{7}$$

We now derive (7) from (1) (with $g_\phi^\dagger$ in place of $g_\phi^{-1}$). By the chain rule, we have:

$$\mathbf{J}[g_\phi]\left(g_\phi^\dagger(x)\right) = \mathbf{J}[f_\theta]\left(f_\theta^\dagger(x)\right) \mathbf{J}[h_\eta]\left(g_\phi^\dagger(x)\right). \tag{8}$$

The Jacobian-transpose Jacobian term in (1) thus becomes:

$$\left| \det \mathbf{J}[g_\phi]^\top \left( g_\phi^\dagger(x) \right) \mathbf{J}[g_\phi] \left( g_\phi^\dagger(x) \right) \right|^{-1/2} \tag{9}$$

$$= \left| \det \mathbf{J}[h_\eta]^\top \left( g_\phi^\dagger(x) \right) \mathbf{J}[f_\theta]^\top \left( f_\theta^\dagger(x) \right) \mathbf{J}[f_\theta] \left( f_\theta^\dagger(x) \right) \mathbf{J}[h_\eta] \left( g_\phi^\dagger(x) \right) \right|^{-1/2}$$

$$= \left| \det \mathbf{J}[h_\eta]^\top \left( g_\phi^\dagger(x) \right) \right|^{-1/2} \left| \det \mathbf{J}[f_\theta]^\top \left( f_\theta^\dagger(x) \right) \mathbf{J}[f_\theta] \left( f_\theta^\dagger(x) \right) \right|^{-1/2} \left| \det \mathbf{J}[h_\eta] \left( g_\phi^\dagger(x) \right) \right|^{-1/2}$$

$$= \left| \det \mathbf{J}[h_\eta] \left( g_\phi^\dagger(x) \right) \right|^{-1} \left| \det \mathbf{J}[f_\theta]^\top \left( f_\theta^\dagger(x) \right) \mathbf{J}[f_\theta] \left( f_\theta^\dagger(x) \right) \right|^{-1/2},$$

where the second equality follows from the fact that $\mathbf{J}[h_\eta]^\top(g_\phi^\dagger(x))$, $\mathbf{J}[f_\theta]^\top(f_\theta^\dagger(x))\mathbf{J}[f_\theta](f_\theta^\dagger(x))$, and $\mathbf{J}[h_\eta](g_\phi^\dagger(x))$ are all square $d \times d$ matrices; and the third equality follows because determinants are invariant to transpositions. The observation that the three involved matrices are square is the reason behind why we can decompose the change-of-variable formula for $g_\phi$ as applying first the change-of-variable formula for $h_\eta$, and then applying it for $f_\theta$.

This property, unlike in the case of standard flows, does not always hold. That is, the change-of-variable formula for a composition of injective transformations is not necessarily equivalent to applying the injective change-of-variable formula twice. To see this, consider the case where $g_1 : \mathbb{R}^d \to \mathbb{R}^{d_2}$ and $g_2 : \mathbb{R}^{d_2} \to \mathbb{R}^D$ are injective, where $d < d_2 < D$ and let $g = g_2 \circ g_1$. Clearly $g$ is injective by construction, and thus the determinant from its change-of-variable formula at a point $z \in \mathbb{R}^d$ is given by:

$$\det \mathbf{J}[g]^\top(z)\mathbf{J}[g](z) = \det \mathbf{J}[g_1]^\top(z)\mathbf{J}[g_2]^\top(g_1(z)) \mathbf{J}[g_2](g_1(z)) \mathbf{J}[g_1](z), \tag{10}$$

where now $\mathbf{J}[g_1](z) \in \mathbb{R}^{d_2 \times d}$ and $\mathbf{J}[g_2](g_1(z)) \in \mathbb{R}^{D \times d_2}$. Unlike the determinant from (9), this determinant cannot be easily decomposed into a product of determinants since the involved matrices are not all square. In particular, (10) need not match:

$$\det \mathbf{J}[g_1]^\top(z)\mathbf{J}[g_1](z) \cdot \det \mathbf{J}[g_2]^\top(g_1(z))\mathbf{J}[g_2](g_1(z)), \tag{11}$$

which would be the determinant terms from applying the change-of-variable formula twice. Note that this observation does not imply that a flow like $g$ could not be trained with our method, it simply implies that the $\det \mathbf{J}[g]^\top(z)\mathbf{J}[g](z)$ term has to be considered as a whole, and not decomposed into separate terms. It is easy to verify that in general, only an initial $d$-dimensional square flow can be separated from the overall Jacobian-transpose-Jacobian determinant.

## C. Full Hutchinson-Based Objective

Here, we provide the full contribution of a point $x$ to the objective containing Hutchinson's estimator and conjugate gradients:

$$\log p_Z \left( g_\phi^\dagger(x) \right) - \log \left| \det \mathbf{J}[h_\eta] \left( g_\phi^\dagger(x) \right) \right| - \beta \left\| x - f_\theta \left( f_\theta^\dagger(x) \right) \right\|_2^2 \tag{12}$$

$$- \frac{1}{2K} \sum_{k=1}^K \texttt{stop\_gradient} \left( \texttt{CG} \left( J_\theta^\top J_\theta; \epsilon_k \right)^\top \right) J_\theta^\top J_\theta \epsilon_k.$$

## D. Conjugate Gradients

We outline the CG algorithm in Algorithm 1, whose output we write as $\texttt{CG}(A; \epsilon)$ in the main manuscript. Note that CG does not need access to $A$, just a matrix-vector product routine against $A$, $\texttt{mvp\_A}(\cdot)$. If $A$ is symmetric positive definite, then CG converges in at most $d$ steps, i.e. its output matches $A^{-1}\epsilon$ and the corresponding residual is 0, and CG uses thus at most $d$ calls to $\texttt{mvp\_A}(\cdot)$. This convergence holds mathematically, but can be violated numerically if $A$ is ill-conditioned, which is why the $\tau < d$ condition is added in the while loop.

---

**Algorithm 1** CG

---

**Input** : mvp_A$(\cdot)$, function for matrix-vector products against $A \in \mathbb{R}^{d \times d}$
$\quad\quad\quad \epsilon \in \mathbb{R}^d$
$\quad\quad\quad \delta \geq 0$, tolerance
**Output** : $A^{-1}\epsilon$
$u_0 \leftarrow \mathbf{0} \in \mathbb{R}^d$ // current solution
$r_0 \leftarrow -\epsilon$ // current residual
$q_0 \leftarrow r_0$
$\tau \leftarrow 0$
**while** $||r_\tau||_2 > \delta$ **and** $\tau < d$ **do**
$\quad\quad v_\tau \leftarrow$ mvp_A$(q_\tau)$
$\quad\quad \alpha_\tau \leftarrow (r_\tau^\top r_\tau)/(q_\tau^\top v_\tau)$
$\quad\quad u_{\tau+1} \leftarrow u_\tau + \alpha_\tau q_\tau$
$\quad\quad r_{\tau+1} \leftarrow r_\tau - \alpha_\tau v_\tau$
$\quad\quad \beta_\tau \leftarrow (r_{\tau+1}^\top r_{\tau+1})/(r_\tau^\top r_\tau)$
$\quad\quad q_{\tau+1} \leftarrow r_{\tau+1} + \beta_\tau q_\tau$
$\quad\quad \tau \leftarrow \tau + 1$

**end**
**return** $u_\tau$

---

# E. Automatic Differentiation

Here we summarize the relevant properties from forward- and backward-mode automatic differentiation (AD) which we use in the main manuscript. Let $f$ be the composition of smooth functions $f_1, \ldots, f_L$, i.e. $f = f_L \circ f_{L-1} \circ \cdots \circ f_1$. For example, in our setting this function could be $f_\theta$, so that $f_1 = \texttt{pad}$, and the rest of the functions could be coupling layers from a $D$-dimensional square flow (or the functions whose compositions results in the coupling layers). By the chain rule, the Jacobian of $f$ is given by:

$$\mathbf{J}[f](z) = \mathbf{J}[f_L](\bar{f}_{L-1}(z)) \cdots \mathbf{J}[f_2](\bar{f}_1(z)) \mathbf{J}[f_1](z), \tag{13}$$

where $\bar{f}_l := f_l \circ f_{l-1} \circ \cdots \circ f_1$ for $l = 1, 2, \ldots, L-1$. Forward-mode AD computes products from right to left, and is thus efficient for computing jvp operations. Computing $\mathbf{J}[f](z)\epsilon$ is thus obtained by performing $L$ matrix-vector multiplications, one against each of the Jacobians on the right hand side of (13). Backward-mode AD computes products from left to right, and would thus result in significantly more inefficient jvp evaluations involving $L - 1$ matrix-matrix products, and a single matrix-vector product. Analogously, backward-mode AD computes vjps of the form $v^\top \mathbf{J}[f](z)$ efficiently, using $L$ vector-matrix products, while forward-mode AD would require $L - 1$ matrix-matrix products and a single vector-matrix product.

Typically, the cost of evaluating a matrix-vector or vector-matrix product against $\mathbf{J}[f_{l+1}](\bar{f}_l)$ (or $\mathbf{J}[f_1](z)$) is the same as computing $\bar{f}_{l+1}(z)$ from $\bar{f}_l(z)$, i.e. the cost of evaluating $f_{l+1}$ (or the cost of evaluating $f_1$ in the case of $\mathbf{J}[f_1](z)$) (Baydin et al., 2018). jvp and vjp computations thus not only have the same computational cost, but this cost is also equivalent to a forward pass, i.e. computing $f$.

When computing $f$, obtaining a jvp with forward-mode AD adds the same memory cost as another computation of $f$ since intermediate results do not have to be stored. That is, in order to compute $\mathbf{J}[f_l](\bar{f}_{l-1}(z)) \cdots \mathbf{J}[f_1](z)\epsilon$, we only need to store $\mathbf{J}[f_{l-1}](\bar{f}_{l-2}(z)) \cdots \mathbf{J}[f_1](z)\epsilon$ and $\bar{f}_{l-1}(z)$ (which has to be stored anyway for computing $f$) in memory. On the other hand, computing a vjp with backward-mode AD has a higher memory cost: One has to first compute $f$ and store all the intermediate $\bar{f}_l(z)$ (along with $z$), since computing $v^\top \mathbf{J}[f_L](\bar{f}_{L-1}(z)) \cdots \mathbf{J}[f_l](\bar{f}_{l-1}(z))$ from $v^\top \mathbf{J}[f_L](\bar{f}_{L-1}(z)) \cdots \mathbf{J}[f_{l+1}](\bar{f}_l(z))$ requires having $\bar{f}_{l-1}(z)$ in memory.

# F. Batch Normalization

We now explain the issues that arise when combining batch normalization with vjps. These issues arise not only in our setting, but every time backward-mode AD has to be called to compute or approximate the gradient of the determinant term.

We consider the case with a batch of size 2, $x_1$ and $x_2$, as it exemplifies the issue and the notation becomes simpler. Consider applying $f_\theta$ (without batch normalization) to each element in the batch, which we denote with the batch function $F_\theta$:

$$F_\theta(x_1, x_2) := (f_\theta(x_1), f_\theta(x_2)). \tag{14}$$

The Jacobian of $F_\theta$ clearly has a block-diagonal structure:

$$\mathbf{J}[F_\theta](x_1, x_2) = \begin{pmatrix} \mathbf{J}[f_\theta](x_1) & \mathbf{0} \\ \mathbf{0} & \mathbf{J}[f_\theta](x_2) \end{pmatrix}. \tag{15}$$

This structure implies that relevant computations such as `vjps`, `jvps`, and determinants parallelize over the batch:

$$(v_1, v_2)^\top \mathbf{J}[F_\theta](x_1, x_2) = \left( v_1^\top \mathbf{J}[f_\theta](x_1), v_2^\top \mathbf{J}[f_\theta](x_2) \right) \tag{16}$$

$$\mathbf{J}[F_\theta](x_1, x_2) \begin{pmatrix} \epsilon_1 \\ \epsilon_2 \end{pmatrix} = \begin{pmatrix} \mathbf{J}[f_\theta](x_1)\epsilon_1 \\ \mathbf{J}[f_\theta](x_2)\epsilon_2 \end{pmatrix}$$

$$\det \mathbf{J}[F_\theta]^\top(x_1, x_2)\mathbf{J}[F_\theta](x_1, x_2) = \det \mathbf{J}[f_\theta]^\top(x_1)\mathbf{J}[f_\theta](x_1) \det \mathbf{J}[f_\theta]^\top(x_2)\mathbf{J}[f_\theta](x_2).$$

In contrast, when using batch normalization, the resulting computation $F_\theta^{BN}(x_1, x_2)$ does not have a block-diagonal Jacobian, and thus this parallelism over the batch breaks down, in other words:

$$(v_1, v_2)^\top \mathbf{J}\left[F_\theta^{(BN)}\right](x_1, x_2) \neq \left( v_1^\top \mathbf{J}[f_\theta](x_1), v_2^\top \mathbf{J}[f_\theta](x_2) \right) \tag{17}$$

$$\mathbf{J}\left[F_\theta^{BN}\right](x_1, x_2) \begin{pmatrix} \epsilon_1 \\ \epsilon_2 \end{pmatrix} \neq \begin{pmatrix} \mathbf{J}[f_\theta](x_1)\epsilon_1 \\ \mathbf{J}[f_\theta](x_2)\epsilon_2 \end{pmatrix}$$

$$\det \mathbf{J}\left[F_\theta^{BN}\right]^\top(x_1, x_2)\mathbf{J}\left[F_\theta^{BN}\right](x_1, x_2) \neq \det \mathbf{J}[f_\theta]^\top(x_1)\mathbf{J}[f_\theta](x_1) \det \mathbf{J}[f_\theta]^\top(x_2)\mathbf{J}[f_\theta](x_2),$$

where the above $\neq$ signs should be interpreted as "not generally equal to" rather than always not equal to, as equalities could hold coincidentally in rare cases.

In square flow implementations, AD is never used to obtain any of these quantities, and the Jacobian log determinants are explicitly computed for each element in the batch. In other words, this batch dependence is ignored in square flows, both in the log determinant computation, and when backpropagating through it. Elaborating on this point, AD is only used to backpropagate (with respect to $\theta$) over this explicit computation. If AD was used on $F_\theta^{BN}$ to construct the matrices and we then computed the corresponding log determinants, the results would not match with the explicitly computed log determinants: The latter would be equivalent to using batch normalization with a `stop_gradient` operation *with respect to $(x_1, x_2)$ but not with respect to $\theta$*, while the former would use no `stop_gradient` whatsoever. Unfortunately, this partial `stop_gradient` operation only with respect to inputs but not parameters is not available in commonly used AD libraries. While our custom implementation of `jvps` can be easily "hard-coded" to have this behaviour, doing so for `vjps` would require significant modifications to PyTorch. We note that this is *not* a fundamental limitation and that these modifications could be done to obtain `vjps` that behave as expected with a low-level re-implementation of batch normalization, but these fall outside of the scope of our paper. Thus, in the interest of performing computations in a manner that remains consistent with what is commonly done for square flows and that allows fair comparisons of our exact and stochastic methods, we avoid using batch normalization.

## G. FID and FID-like Scores

For a given dataset $\{x_1, \ldots, x_n\} \subset \mathbb{R}^D$ and a set of samples generated by a model $\{x_1^{(g)}, \ldots, x_m^{(g)}\} \subset \mathbb{R}^D$, along with a statistic $T : \mathbb{R}^D \to \mathbb{R}^r$, the empirical means and covariances are given by:

$$\hat{\mu} := \frac{1}{n} \sum_{i=1}^n T(x_i), \qquad \hat{\Sigma} := \frac{1}{n-1} \sum_{i=1}^n \left( T(x_i) - \hat{\mu} \right) \left( T(x_i) - \hat{\mu} \right)^\top \tag{18}$$

$$\hat{\mu}^{(g)} := \frac{1}{m} \sum_{i=1}^m T\left( x_i^{(g)} \right), \quad \hat{\Sigma}^{(g)} := \frac{1}{m-1} \sum_{i=1}^m \left( T\left( x_i^{(g)} \right) - \hat{\mu}^{(g)} \right) \left( T\left( x_i^{(g)} \right) - \hat{\mu}^{(g)} \right)^\top. \tag{19}$$

*Table 3.* Training times in seconds, "$K > 1$" means $K = 10$ for tabular data and $K = 4$ for images.

| Dataset | RNFs-ML (exact) | | RNFs-ML ($K = 1$) | | RNFs-ML ($K > 1$) | | RNFs-TS | |
|---|---|---|---|---|---|---|---|---|
| | EPOCH | TOTAL | EPOCH | TOTAL | EPOCH | TOTAL | EPOCH | TOTAL |
| POWER | 53.8 | 4.13e3 | 67.4 | 6.76e3 | 136 | 1.14e4 | 45.1 | 3.83e3 |
| GAS | 37.3 | 2.51e3 | 62.7 | 4.51e3 | 80.1 | 5.24e3 | 43.2 | 3.49e3 |
| HEPMASS | 143 | 1.01e4 | 146 | 8.28e3 | 159 | 1.20e4 | 29.1 | 2.42e3 |
| MINIBOONE | 49.3 | 4.16e3 | 26.3 | 2.01e3 | 29.8 | 2.94e3 | 4.61 | 481 |
| MNIST | 2.40e3 | 2.59e5 | 1.71e3 | 1.57e5 | 3.03e3 | 3.20e5 | 2.13$e$2 | 3.90e4 |
| FMNIST | 2.34e3 | 2.59e5 | 1.72e3 | 1.50e5 | 3.15e3 | 2.10e5 | 1.04e2 | 1.11e4 |

The FID score takes $T$ as the last hidden layer of a pretrained inception network (Szegedy et al., 2015), and evaluates generated sample quality by comparing generated moments against data moments. This comparison is done with the squared Wasserstein-2 distance between Gaussians with corresponding moments, which is given by:

$$\left|\left|\hat{\mu} - \hat{\mu}^{(g)}\right|\right|_2^2 + \text{tr}\left(\hat{\Sigma} + \hat{\Sigma}^{(g)} - 2\left(\hat{\Sigma}\hat{\Sigma}^{(g)}\right)^{1/2}\right), \tag{20}$$

which is 0 if and only if the moments match. Our proposed FID-like score for tabular data is computed the exact same way, except no inception network is used. Instead, we simply take $T$ to be the identity, $T(x) = x$.

## H. Experimental Details

First we will comment on hyperparameters/architectural choices shared across experiments. The $D$-dimensional square flow that we use, as mentioned in the main manuscript, is a RealNVP network (Dinh et al., 2017). In all cases, we use the ADAM (Kingma & Ba, 2015) optimizer and train with early stopping against some validation criterion specified for each experiment separately and discussed further in each of the relevant subsections below. We use no weight decay. We also do not use batch normalization in any experiments for the reasons mentioned above in Appendix F. We use a standard Gaussian on $d$ dimensions as $p_Z$ in all experiments.

**Compute** We ran our two-dimensional experiments on a Lenovo T530 laptop with an Intel i5 processor, with negligible training time per epoch. We ran the tabular data experiments on a variety of NVIDIA GeForce GTX GPUs on a shared cluster: we had, at varying times, access to 1080, 1080 Ti, and 2080 Ti models, but never access to more than six cards in total at once. For the image experiments, we had access to a 32GB-configuration NVIDIA Tesla v100 GPU. We ran each of the tabular and image experiments on a single card at a time, except for the image experiments for the RNFs-ML (exact) and ($K = 10$) models which we parallelized over four cards.

Table 3 includes training times for all of our experiments. Since we used FID-like and FID scores for ealy stopping, we include both per-epoch and total times. Per epoch times of RNFs-ML exclude epochs where the Jacobian-transpose-Jacobian log determinant is annealed with a 0 weight, although we include time added from this portion of training into the total time cost. Note throughout this section we also consider one epoch of the two-step baseline procedure to be one full pass through the data training the likelihood term, and then one full pass through the data training the reconstruction term.

### H.1. Simulated Data

The data for this experiment is simulated from a von Mises distribution centred at $\frac{\pi}{2}$ projected onto a circle of radius 1. We randomly generate 10,000 training data points and train with batch sizes of 1,000. We use 1,000 points for validation, performing early stopping using the value of the full objective and halting training when we do not see any validation improvement for 50 epochs. We create visualizations in Figure 2 and Figure 3 by taking 1,000 grid points equally-spaced between $-3$ and 3 as the low-dimensional space, project these into higher dimensions by applying the flow $g_\phi$, and then assign density to these points using the injective change-of-variable formula (1). In this low-dimensional example, we use the full Jacobian-transpose-Jacobian which ends up just being a scalar as $d = 1$. We commence likelihood annealing (when active) on the 500-th training epoch and end up with a full likelihood term by the 1000-th.

For the $D$-dimensional square flow $f_\theta$, we used a 5-layer RealNVP model, with each layer having a fully-connected coupler network of size $2 \times 10$, i.e. 2 hidden layers each of size 10, outputting the shift and (log) scale values. The baseline

additionally uses a simple shift-and-scale transformation in $d$-dimensional space as $h_\eta$; we simply use the identity map for $h_\eta$ in this simple example.

We perform slightly different parameter sweeps for the two methods based on preliminary exploration. For the baseline two-step procedure, we perform runs over the following grid:

- Learning rate: $10^{-3}$, $10^{-4}$.

- Regularization parameter ($\beta$): 100, 1,000, 10,000 (which for this method is equivalent to having a separate learning rate for the regularization objective).

- Likelihood annealing: `True` or `False`.

For our method, we search over the following, excluding learning rate since our method was stable at the higher rate of $10^{-3}$:

- Regularization parameter ($\beta$): 10, 50, 200.

- Likelihood annealing: `True` or `False`.

Empirically we found the two-step baseline performed better with the higher regularization, which also agrees with the hyperparameter settings from their paper.

**Divergences on RNFs-TS between our codebase and the implementation of (Brehmer & Cranmer, 2020)**   Although we were able to replicate the baseline RNF-TS method, there were some different choices made in the codebase of the baseline method (available here: https://github.com/johannbrehmer/manifold-flow), which we outline below:

- The baseline was trained for 120 epochs and then selects the model with best validation score, whereas we use early stopping over an (essentially) unlimited number of epochs.

- The baseline weights the reconstruction term with a factor of 100 and the likelihood term with a factor of 0.1. This is equivalent in our codebase to setting $\beta = 1{,}000$, and lowering the learning rate by a factor of 10.

- The baseline uses cosine annealing of the learning rate, which we do not use.

- The baseline includes a sharp Normal base distribution on the pulled-back padded coordinates. We neglected to include this as it isn't mentioned in the paper and can end up resulting in essentially a square flow construction.

- The baseline uses the ADAMW optimizer (Loshchilov & Hutter, 2019) to fix issues with weight decay within ADAM (which they also use). We stick with standard ADAM as we do not use weight decay.

- The baseline flow reparametrizes the scale $s$ of the RealNVP network as $s = \sigma(\tilde{s} + 2) + 10^{-3}$, where $\tilde{s}$ is the unconstrained scale and $\sigma$ is the sigmoid function, but this constrains the scale to be less than $1 + 10^{-3}$. This appears to be done for stability of the transformation (cf. the ResNets below). We instead use the standard parametrization of $s = \exp(\tilde{s})$ as the fully-connected networks appear to be adequately stable.

- The Baseline uses ResNets with ReLU activation of size $2 \times 100$ as the affine coupling networks. We use MLPs with tanh activation function instead.

- The baseline uses a dataset which is not strictly on a manifold. The radius of a point on the circle is sampled from $\mathcal{N}(1, 0.01^2)$. We use a strictly one-dimensional distribution instead with a von Mises distribution on the angle as noted above.

In general, we favoured more standard and simpler choices for modelling the circle, outside of the likelihood annealing which is non-standard.

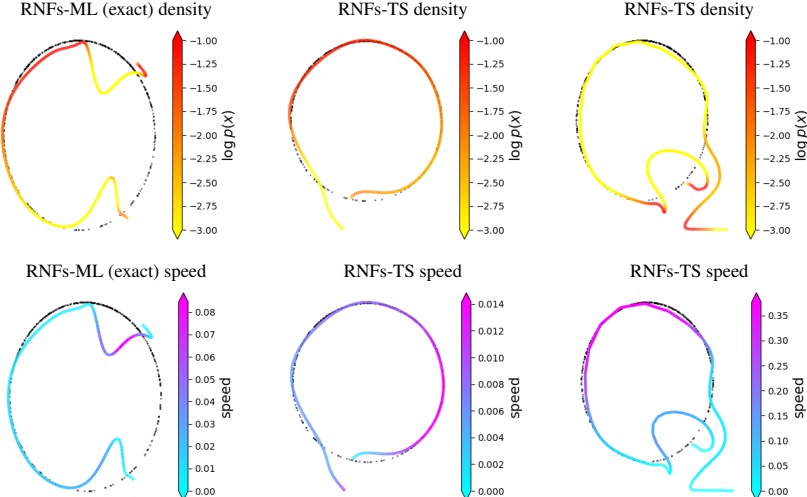

Figure 3. Densities (top row) and speeds (bottom row) for additional runs. Failed runs not recovering neither manifold nor the distribution on it, RNFs-ML (exact) (left column) and RNFs-TS (right column). Successful RNFs-TS run (middle column).

We note that, while the results reported in the main manuscript are representative of common runs, both for RNFs-ML (exact) and RNFs-TS; not every single run of RNFs-ML (exact) obtained results as good as the ones from the main manuscript. Similarly, some runs of RNFs-TS recovered better likelihoods than the one from the main manuscript. We emphasize again that the results reported on the main manuscript are the most common ones: most RNFs-ML (exact) runs correctly recovered both the manifold and the distribution on it, and most RNFs-TS runs recovered only the manifold correctly. For completeness, we include in Figure 3 some of the rare runs where results were different than the ones reported in the main manuscript. Interestingly, we can see that the successful RNFs-TS run, which managed to recover the distribution on the manifold, had more constant speeds than other RNFs-TS runs.

### H.2. Tabular Data

For the tabular data, we use the GAS, POWER, HEPMASS, and MINIBOONE datasets, preprocessed as in Papamakarios et al. (2017), although we neglect to use a test dataset as we simply compared moments on the trained data, as is typically done with the FID score. We did not observe problems with overfitting in practice for any of the methods. We use the FID-like metric with the first and second moments of the generated and observed data as described in Appendix G for early stopping, halting training after 20 epochs of no improvement.

We again use a RealNVP flow in $D$ dimensions but now with 10 layers, with each layer having a fully-connected coupler network of hidden dimension $4 \times 128$. The $d$-dimensional flow here is also a RealNVP, but just a 5-layer network with couplers of size $2 \times 32$.

In all methods, we use a regularization parameter of $\beta = 50$. We introduce the likelihood term with low weight after 25 epochs, linearly increasing its contribution to the objective until it is set to its full weight after 50 epochs. We select $d$ as $\lfloor \frac{D}{2} \rfloor$, except for ML methods on $D = 8$ GAS which use $d = 2$ (noted below). We use a learning rate of $10^{-4}$. For the methods involving the Hutchinson estimator, we use a standard Gaussian as the estimating distribution. We also experimented with a Rademacher distribution here but found the Gaussian to be superior.

Results reported on the main manuscript are the mean of 5 runs (with different seeds) plus/minus standard error. Occasionally, both RNFs-ML and RNFs-TS resulted in failed runs with FID-like scores at least an order of magnitude larger than other runs. In these rare instances, we did another run and ignored the outlier. We did this for both methods, and we do point out that RNFs-ML did not have a higher number of failed runs.

As mentioned in the main manuscript, GAS required slightly more tuning as RNFs-ML did not outperform RNFs-TS when using $d = 4$. We instead use latent dimension $d = 2$, where this time RNFs-ML did outperform. Since RNFs-TS did better with $d = 4$, we report those numbers in the main manuscript. Otherwise, our methods outperformed the baseline out-of-the-box, using parameter configurations gleaned from the image and circle experiments.

*Table 4.* Parameter combinations investigated for MNIST runs. Note that the final two rows are irrelevant for RNF-ML (exact) and RNF-TS. We include "short names" for ease of listing parameters for the runs in Table 2.

| PARAMETER | SHORT NAME | MAIN VALUE | ALTERNATIVES |
|---|---|---|---|
| Likelihood Annealing | LA | True | False |
| Reconstruction parameter | $\beta$ | 50 | $5, 500, 10000$ |
| Low dimension | $d$ | 20 | $10, 15, 30$ |
| $D$-dim flow coupler | $D$ NET | $8 \times 64$ | $4 \times 64$ |
| $d$-dim flow layers | $d$ LAYERS | 5 | 10 |
| Hutchinson distribution | HUTCH | Gaussian | Rademacher |
| CG tolerance (normalized) | `tol` | 1 | 0.001 |

*Table 5.* Parameter choices for the MNIST runs reported in Table 2.

| METHOD | LA | $\beta$ | $d$ | $D$ NET | $d$ LAYERS | HUTCH | `tol` |
|---|---|---|---|---|---|---|---|
| RNFs-ML (exact) | True | 5 | 20 | $8 \times 64$ | 10 | N/A | N/A |
| RNFs-ML ($K = 1$) | True | 5 | 20 | $8 \times 64$ | 10 | Gaussian | 0.001 |
| RNFs-ML ($K = 4$) | True | 50 | 20 | $8 \times 64$ | 5 | Gaussian | 1 |
| RNFs-TS | True | 50 | 20 | $8 \times 64$ | 5 | N/A | N/A |

### H.3. Image Data and Out-of-Distribution Detection

In this set of experiments, we mostly tuned the RNFs-ML methods on MNIST for $K = 1$ – applying any applicable settings to RNFs-TS on MNIST as well – which is likely one of the main reasons that RNFs-ML perform so well for $K = 1$ vs. the exact method or $K = 4$. The reason why we spent so much time on $K = 1$ is that it was the fastest experiment to run and thus the easiest to iterate on. Our general strategy for tuning was to stick to a base set of parameters that performed reasonably well and then try various things to improve performance. A full grid search of all the parameters we might have wanted to try was quite prohibitive on the compute that we had available. Some specific details on settings follow below.

For the $D$-dimensional square flow, we mainly used the 10-layer RealNVP model which exactly mirrors the setup that Dinh et al. (2017) used on image data, except we neglect to include batch normalization (as discussed in Appendix F) and we also tried reducing the size of the ResNet coupling networks from $8 \times 64$ to $4 \times 64$ for computational purposes. For further computational savings, we additionally attempted to use a RealNVP with fewer layers as the $D$-dimensional square flow, but this performed extremely poorly and we did not revisit it. For the $d$-dimensional square component, we used another RealNVP with either 5 or 10 layers, and fully-connected coupler networks of size $4 \times 32$. We also looked into modifying the flow here to be a neural spline flow (Durkan et al., 2019), but this, like the smaller $D$-dimensional RealNVP, performed very poorly as well. This may be because we did not constrain the norm of the gradients, although further investigation is required. We also looked into using no $d$-dimensional flow for our methods as in the circle experiment, but this did not work well at all.

For padding, we first randomly (although this is fixed once the run begins) permute the $d$-dimensional input, pad to get to the appropriate length of vector, and then reshape to put into image dimension. We also pad with zeros when performing the inverse of the density split operation (cf. the $z$ to $x$ direction of Dinh et al. (2017, Figure 4(b))), so that the input is actually padded twice at various steps of the flow.

When we used likelihood annealing, we did the same thing as for the tabular data: optimize only the reconstruction term for 25 epochs, then slowly and linearly introduce the likelihood term up until it has a weight of 1 in the objective function after epoch 50.

We summarize our attempted parameters in Table 4. For some choices of parameters, such as likelihood annealing set to `False`, $d = 15, 30$, $\beta = 10,000$, and CG tolerance set to 1, we had very few runs because of computational reasons. However, we note that the run with low CG tolerance ends up being the most successful run on MNIST. We have included "SHORT NAMES" in the table for ease of listing hyperparameter values for the runs in Table 2, which we now provide for MNIST and FMNIST in Table 5 and Table 6 respectively.

Figure 4 shows best RNFs-ML log-likelihoods for models trained on MNIST (**left panel**), and we can see that indeed

*Table 6.* Parameter choices for the FMNIST runs reported in Table 2.

| METHOD | LA | $\beta$ | $d$ | $D$ NET | $d$ LAYERS | HUTCH | tol |
|--------|-----|---------|-----|---------|-----------|-------|-----|
| RNFs-ML (exact) | True | 50 | 20 | $8 \times 64$ | 10 | N/A | N/A |
| RNFs-ML ($K = 1$) | True | 50 | 20 | $8 \times 64$ | 5 | Rademacher | 1 |
| RNFs-ML ($K = 4$) | True | 50 | 20 | $8 \times 64$ | 10 | Rademacher | 1 |
| RNFs-TS | False | 5 | 20 | $4 \times 64$ | 10 | N/A | N/A |

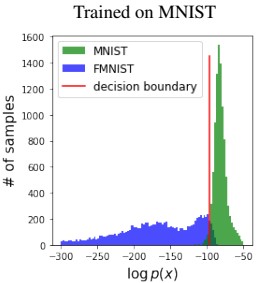 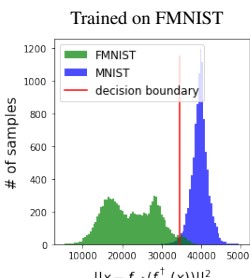 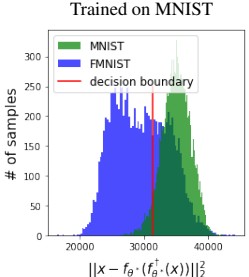

*Figure 4.* OoD log-likelihood histograms trained on MNIST (left), and OoD reconstruction error histograms trained on FMNIST (middle) and MNIST (right). Log-likelihood results (left) are RNFs-ML (exact), and reconstruction results (middle and right) are RNFs-ML ($K = 1$). Note that green denotes in-distribution data, and blue OoD data; and colors *do not* correspond to datasets.

MNIST is assigned higher likelihoods than FMNIST. We also include OoD detection results when using reconstruction error instead of log-likelihoods, for models trained on FMNIST (**middle panel**) and MNIST (**right panel**). We observed similar results with RNFs-TS. Surprisingly, it is now the reconstruction error which exhibits puzzling behaviour: it is always lower on FMNIST, regardless of whether the model was trained on FMNIST or MNIST. Once again, this behaviour also happens for RNFs-TS, where the reconstruction error is optimized separately. We thus hypothesize that this behaviour is not due to maximum likelihood training, and rather is a consequence of inductive biases of the architecture.