# OpenReview forum: "Rectangular Flows for Manifold Learning"
_ICML.cc/2021/Workshop/INNF — INNF+ 2021 spotlighttalk_

### Official Review · Reviewer_7gXk · 2021-06-11

**Rating:** Accept
**Confidence:** 3

**Summary:**

The paper introduces a new method for training and evaluating normalizing flows where the latent variable is of lower dimension than the observable. These are termed rectangular flows (as the Jacobian is no longer a square matrix). The difficulty is in evaluating the gradient of the product of the transposed Jacobian with the Jacobian, in the change-of-variable formula. The authors introduce two strategies. The first is an unbiased gradient estimator based on Hutchinson's trace estimator with the inverse computed using the method of conjugate gradients. The second, more precise at a greater computational cost, relies on the computation of vector/Jacobian products through reverse-mode AD and Jacobian/vector products throug forward mode AD.

The method is evaluated on synthetic data, tabular data and on FMNIST/MNIST OOD detection, and provides convincing performance compared to baselines.

**Justification For Rating:**

The method solves a clearly defined problem in a convincing way. The write-up is very precise and easy to follow and the evaluation is more than sufficient to show that the approach is worth considering.

It is a little mysterious to me exactly how the MNIST/FMNIST experiment works. For instance, how is the decision boundary arrived at and if, as the authors emphasize, the colors do not correspond to datasets, what do they correspond to. A short step-by step walkthrough of the experiment would be illustrative. If possible, this experiment should be repeated on other MNIST-like  data to isolate any effects that are particular to MNIST, but I think what is presented here is sufficient for the venue.

---

### Official Review · Reviewer_sWMz · 2021-06-12

**Rating:** Accept
**Confidence:** 4

**Summary:**

This paper addresses the shortcomings to injective flows, a method to learn flows on a low-dimensional manifold. Current injective flows assume the manifold is known or avoid the volume-change term. This work proposes a tractable injective flow method that makes no assumption on the manifold and computer the volume-change term with some estimators.


**Justification For Rating:**

This work overcomes two key limitations to current work on injective flows by directly computing the intractable term in the optimization objective with the use of Huthinson's trace, Conjugate Gradients, etc. Additionally, an exact method is derived which avoids the Hutchinson estimator. This is important as it presents a much more general approach to injective flows. It is also shown to be competitive compared to recent work.

The exact method uses some automatic differentiation tricks but at the expense of memory. It would be great to see this work expanded to more modern flow architectures instead of RealNVP. And crucially, will flows on manifold still be tractable with powerful architectures such as Residual Flows?

---

### Official Review · Reviewer_C6wg · 2021-06-12

**Rating:** Accept
**Confidence:** 3

**Summary:**

The authors propose a method that allows them to simultaneously learn a manifold and a density on it using maximum likelihood. In particular, building on previous work of Brehmer & Cranmer (2020), they consider injective flows that allow for learning a distribution supported on a low-dimensional manifold. This paper addresses one of the key challenges identified in the above mentioned work, namely the evaluation of the volume-change term, which is computationally expensive. While Brehmer & Cranmer circumvent this issue by splitting the training objective, the authors of the current work argue that the objective function can be evaluated directly, and they demonstrate that this can yield accurate results in situations where the original method fails.

**Justification For Rating:**

This paper is very clear and well written and addresses a problem highlighted in the recent work of Brehmer & Cranmer,  namely the efficient evaluation of the volume-change term that arises in likelihood training with injective flows. The authors of the current work show that this term can be evaluated efficiently for problems of nearly 1000 dimensions.

However, apart from the computational challenges related to the volume-change term, there is another point which I believe would deserve a more detailed explanation. Brehmer & Cranmer argue that maximum likelihood training is not enough (see Sec. 3) as it does not incentivise the model to learn the correct density because of the projection. The computational challenge of evaluating the volume-change term is highlighted as a separate issue. At the beginning of Sec. 3.1 of the current work, the authors re-iterate this argument and propose to maximise the Lagrangian in eq. 3 instead. However, it's not entirely clear to me how exactly this objective fixes the above mentioned issue and I would find it very helpful if this point was explained in more detail.

---

### Decision · Program_Chairs · 2021-06-14

Accept (spotlight talk)